# Risk of Adverse Drug Events Following the Virtual Addition of COVID-19 Repurposed Drugs to Drug Regimens of Frail Older Adults with Polypharmacy

**DOI:** 10.3390/jcm9082591

**Published:** 2020-08-10

**Authors:** Sweilem B. Al Rihani, Matt K. Smith, Ravil Bikmetov, Malavika Deodhar, Pamela Dow, Jacques Turgeon, Veronique Michaud

**Affiliations:** 1Tabula Rasa HealthCare Precision Pharmacotherapy Research & Development Institute, Orlando, FL 32827, USA; srihani@trhc.com (S.B.A.R.); mksmith@trhc.com (M.K.S.); rbikmetov@trhc.com (R.B.); mdeodhar@trhc.com (M.D.); pdow@trhc.com (P.D.); jturgeon@trhc.com (J.T.); 2Faculty of Pharmacy, Université de Montréal, Montreal, QC H3C 3J7, Canada

**Keywords:** adverse drug events, COVID-19, polypharmacy, older adults, simulation

## Abstract

Determination of the risk–benefit ratio associated with the use of novel coronavirus disease 2019 (COVID-19) repurposed drugs in older adults with polypharmacy is mandatory. Our objective was to develop and validate a strategy to assess risk for adverse drug events (ADE) associated with COVID-19 repurposed drugs using hydroxychloroquine (HCQ) and chloroquine (CQ), alone or in combination with azithromycin (AZ), and the combination lopinavir/ritonavir (LPV/r). These medications were virtually added, one at a time, to drug regimens of 12,383 participants of the Program of All-Inclusive Care for the Elderly. The MedWise Risk Score (MRS^TM^) was determined from 198,323 drug claims. Results demonstrated that the addition of each repurposed drug caused a rightward shift in the frequency distribution of MRS^TM^ values (*p* < 0.05); the increase was due to an increase in the drug-induced Long QT Syndrome (LQTS) or CYP450 drug interaction burden risk scores. Increases in LQTS risk observed with HCQ + AZ and CQ + AZ were of the same magnitude as those estimated when terfenadine or terfenadine + AZ, used as positive controls for drug-induced LQTS, were added to drug regimens. The simulation-based strategy performed offers a way to assess risk of ADE for drugs to be used in people with underlying medical comorbidities and polypharmacy at risk of COVID-19 infection without exposing them to these drugs.

## 1. Introduction

In December 2019, an unidentified pneumonia was reported in Wuhan, China [1]. In response, the World Health Organization (WHO) declared an epidemiological alert on 31 December 2019 [1,2]. Within weeks, SARS-CoV-2, spread globally causing the coronavirus disease 2019 (COVID-19) pandemic. As of 5 August 2020, COVID-19 has been diagnosed in more than 18,601,795 patients and associated with over 702,045 deaths all over the world [3].

Hydroxychloroquine (HCQ), chloroquine (CQ), and remdesivir have received US Food and Drug Administration (FDA) emergency use authorization (EUA) based on preliminary results from numerous ongoing clinical trials [4,5]. The FDA has recently determined that the statutory criteria for issuance of an EUA for HCQ and CQ were no longer met. Several other medications are also being tested in numerous clinical trials [6]. So far, no repurposed drug, new drug nor vaccine has been approved as a therapy for COVID-19. However, remdesivir and dexamethasone show promise as COVID-19 treatment in severely ill hospitalized patients [7,8], while the search continues for a treatment that decreases symptoms or rate of severe complications in the community.

In addition to the lack of proven efficacy, the safety to use drugs such as HCQ or CQ has been questioned, and their safety when added to an existing drug regimen, in the context of COVID-19, has not been established yet [9]. In particular, cardiac safety issues including abnormal heart rhythms such as QT prolongation have been associated with HCQ, CQ and azithromycin (AZ) [10,11,12,13,14,15,16]. Serious dysrhythmias in patients with COVID-19 who were treated with HCQ or CQ, often in combination with AZ and other drugs that prolong the QTc interval, have been reported [17,18,19,20].

Tabula Rasa HealthCare (TRHC) has developed a proprietary medication risk score, the MedWise Risk Score (MRS^TM^), that uses algorithms to compute risk of adverse drug events (ADE) while considering five medication characteristics [21,22,23]: (1) computation of a drug regimen relative odds ratio for ADE using the FDA pharmacovigilance database (FDA Adverse Event Reporting System (FAERS)) [24], (2) anticholinergic cognitive burden (ACB) [25], (3) sedative burden (SB) [26,27,28], (4) drug-induced Long QT Syndrome (LQTS) burden [29,30], and (5) CYP450 drug interaction burden risk scores [31]. MRS^TM^ is a predictive tool used by healthcare providers to assess patients’ likelihood of an ADE occurring; over 30 million individuals have been risk-stratified using this tool recently to direct pharmacists and health professional interventions for high-risk patients. [32] In addition to its association with the risk of ADE, higher MRS^TM^ is associated with increased medical expenditures, more hospitalization, more emergency visits, and an increased hospital length of stay in patients with comorbidities and polypharmacy [21].

Programs of All-inclusive Care for the Elderly (PACE), organizations funded by the Centers for Medicare and Medicaid Services, provide supportive services to community-dwelling adults, age 55 and older, who require a “nursing home level of care” [33]. TRHC currently provides pharmacy services to about 100 PACE Centers in the United States, servicing close to 13,000 participants. With several chronic conditions, frail PACE participants clearly represent an at-risk population for COVID-19 infection and complication [34]. Furthermore, PACE participants serviced by TRHC receive on average about 12 different medications per day. Obviously, the addition of any new medication to this polypharmacy condition increases the risk of ADE.

The objective of our study was to assess the risk of ADE associated with the addition of the proposed COVID-19 repurposed drugs HCQ and CQ, alone or in combination with azithromycin (AZ), and with the clinically used combination of lopinavir/ritonavir (LPV/r), to the drug regimens of PACE participants. A simulation approach was used to avoid unnecessary exposure to these drugs in patients and to allow a controlled estimation of risk change by the virtual addition of these drugs, one at a time, to their drug regimen. Validation of such an approach could help estimate the risk of ADE for future repurposed drugs to be used in COVID-19 patients.

## 2. Methods

### 2.1. Subjects and Study Design

In this cohort study, drug insurance claims of individuals enrolled in PACE receiving pharmacy services from CareKinesis were used to perform predictive simulation analyses. The last available prescription drug claims in 2019 were used to determine the subjects’ baseline drug regimen (from 1 July 2019 and ending 31 December 2019) and to calculate their baseline MRS^TM^. Participants with no drug claims data available for the specified period of 2019 were excluded. Data elements analyzed were prescribed drugs, dose, age, sex, and medical claims with an International Statistical Classification of Diseases and Related Health Problems 10th Revision (ICD-10)-WHO Version for 2019 for the corresponding period in 2019. For data protection, date of birth was represented as a year value, and age over 90 years old was fixed at 89 years old. All participant-level data were anonymized before being made available for analysis in this study.

This research protocol was reviewed and approved by Biomedical Research Alliance of New York Institutional Review Board (BRANY IRB), an independent review board, prior to study initiation, and a waiver of authorization to use protected health information was granted (protocol #20-12-117-427). The study protocol was registered at the U.S. National Institutes of Health website (http://www.clinicaltrials.gov; NCT04339634).

### 2.2. Medication Risk Score and Simulation Strategy

A medication risk stratification was used to simulate the impact of different COVID-19 repurposed drugs on the MRS™ (see the Patent section of the manuscript for more details) [21,22,23,30]. The total MRS^TM^ and the individual aggregated risk factors (FDA Adverse Event Reporting System (FAERS), anticholinergic cognitive burden (ACB), sedative burden (SB), drug-induced LQTS, and CYP450 drug interaction burden scores) were divided into sub-categories of Low risk, Moderate risk, and High risk. The range of the total MRS^TM^ values is from 0 to 53 (no risk to highest risk). MRS^TM^ risk groups are defined as follows: MRS^TM^ values < 15 are classified into the Low-risk group, MRS^TM^ values ≥15 to <20 into the Moderate-risk group, and MRS^TM^ values ≥ 20 into the High-risk group.

To simulate the effects of a repurposed drug (or drug combination) on the MRS^TM^, a fictitious claim for the tested repurposed drug (or combination) was added to the database for each participant. If a participant was already taking the added drug under study, their daily dosage was set to the proposed dose of this drug for treating COVID-19. Following the addition of a repurposed drug (or drug combination), a new MRS^TM^ was derived for each participant.

A total of five repurposed drugs or drug combinations were tested, including HCQ (400 mg twice daily), HCQ with AZ (400 mg twice daily + 500 mg once daily), CQ (500 mg twice daily), CQ with AZ (500 mg twice daily + 500 mg once daily), and, LPV/r (400 mg twice daily + 100 mg twice daily). In particular, HCQ and CQ can cause QT prolongation and polymorphic ventricular tachycardia (torsade de pointes) [35,36]. With the LQTS risk score being part of the MRS^TM^ [22,30], additional simulations were performed using known negative and positive controls for QT prolongation including fexofenadine and terfenadine, respectively (fexofenadine 180 mg/day; terfenadine 180 mg/day; and terfenadine 180 mg/day + AZ 500 mg/day) [37].

### 2.3. Data Processing and Statistical Analysis

To perform the medication risk stratification, a webservice interface and customized scripts were used. The MRS^TM^ were generated by processing prescribed drug claims using National Drug Codes (NDCs) as drug identifiers. Medication data were extracted and cleaned from errors and inconsistencies through quality and integrity analyses. Since NDCs can also be assigned to non-medications (e.g., medical devices and consumables), active medication data were further filtered to exclude such NDCs. Afterward, active medication data for each participant was filtered based on prescription dates and days’ supply, which include any possible refills.

Descriptive population characteristics were measured, including means, standard deviations, and proportions as appropriate. For comparing the MRS™ and composite individual risk factors before and after addition of repurposed drugs into participants’ drug regimens, the Wilcoxon signed-rank test analysis was performed. To determine the statistical significance of participants moving to a higher risk stratification category (Low-to-Moderate, Low-to-High, or Moderate-to-High), the McNemar test and the Stuart–Maxwell test of marginal homogeneity were utilized (no participants moved to a lower risk score category). To determine if the MRS™ or a risk category were more influenced by one drug than by others, we used a Friedman test, followed by paired comparisons with the Wilcoxon signed-rank test. For all the Wilcoxon signed-rank tests, the ranks of zeros were included in calculating the statistic (implemented as *zero method* = ‘*zsplit*’ in SciPy 1.4.1). For the statistical analysis assessing the difference between female and male groups, the effect size was calculated using the method denoted *f*.

The diseases were identified by finding the ICD-10 codes with the highest number of participants, limiting the ICD-10 codes to the first three digits. When ICD-9 codes were still used, these codes were translated to the appropriate ICD-10 code, if possible.

For statistical significance, we considered *p*-values below 0.05 to be significant. A population’s mean score change of at least 1 point was considered significant, as reported by Bankes et al. [21]. To adjust for multiple comparisons, the Benjamini/Hochberg adjustment was applied. When the *p*-values were too low, a value of *p* < 0.0001 is indicated. Statistical analysis was performed in Python 3.7.6 using the pandas (v. 1.0.1) (open-source software fiscally sponsored by NumFOCUS, Austin, TX, USA), SciPy (v. 1.4.1)) (open-source software fiscally sponsored by NumFOCUS (Austin, TX, USA)), statsmodels (v. 0.11.0) (open-source software sponsored by Google (Menlo Park, CA, USA) and AQR Capital Management (Greenwich, CT, USA)), Matplotlib (v. 3.1.3) (open-source comprehensive library sponsored by NumFOCUS (Austin, TX, USA)), and seaborn packages (v. 0.10.0) (a Python data visualization library created by Michael Waskom (New York, NY, USA) and in R (v. 1.2.5019) (R foundation for statistical computing. RC Team. (Vienna, Austria) with the dplyr, data.table, sqldf, scales, ggplot2, and igraph packages. Microsoft SQL Server (v. 15) was used to manipulate and analyze large datasets.

## 3. Results

### 3.1. Participant Characteristics

In total, 198,323 prescribed drug claims from 12,383 PACE participants were included in our study. Participants’ demographic and clinical characteristics at baseline are described in Table 1. Overall, the mean age of the PACE participants was 76 years (SD ± 10), 67.4% were female, and the average number of prescribed drugs per day in their actual drug regimen was 11.8 (SD ± 5.7). The more frequently prescribed medications are listed in Appendix A. The most common diseases/symptoms observed in our PACE population were hypertension, dyslipidemia, pain, vitamin D deficiency, constipation, type 2 diabetes, gastro-esophageal reflux disorder (GERD), and chronic obstructive pulmonary disease (COPD). More details on the most common diseases with their corresponding prevalence are provided in Appendix A. The number of participants already taking HCQ, AZ, or LPV/r at baseline was 106, 125, and 5 respectively; no participant was taking CQ. At baseline, the mean MRS^TM^ was 14.4 (SD ± 7.7), with 54.4%, 20.1%, and 25.5% of our PACE population having a Low, Moderate, and High MRS^TM^. The average number of prescribed drugs increased in each category of Low, Moderate, and High-risk groups (from 9 to 14 and to 17 drugs/day, respectively).

Concomitant drugs that can potentially interact with the CYP450 metabolic pathway of HCQ, CQ, and LPV/r were investigated: HCQ is metabolized mainly by CYP2C8 (30%) and to a lesser extent by CYP3A4 and CYP2D6 (both isoforms contributing to 15% of its metabolic clearance); CQ is metabolized by CYP2C8 and CYP3A4 (35% and 15% of its metabolic clearance, respectively); lopinavir is mostly metabolized by CYP3A4 (90%); and ritonavir is a strong substrate that competitively inhibits CYP3A4 [38,39,40,41,42,43]. Frequencies of participants already taking drugs metabolized by CYP2C8, CYP3A4, and CYP2D6 are listed in Table 1. CYP3A4 and CYP2D6 represent two minor pathways for elimination of HCQ when considered separately; therefore, drug regimens simultaneously affecting both pathways have been taken into consideration. Common CYP2C8 interacting drugs found in our PACE population were ibuprofen, loperamide, trimethoprim, pioglitazone, primidone, repaglinide, and gemfibrozil (Appendix A). A total of 6.7% of participants were treated with either 1) a CYP2C8 inhibitor or a CYP2C8 high-affinity substrate drug (N = 693) or 2) a CYP2C8 inducer (N = 137); these drugs could significantly modulate HCQ and CQ plasma concentrations (increase or decrease plasma levels, respectively). In our PACE population, 415 (3.4%) participants were taking a drug regimen that could impede all metabolic pathways involved in HCQ disposition (a combination of CYP2C8, CYP3A4, and CYP2D6 significant interacting drugs).

### 3.2. Simulated Effects of Repurposed Drugs for COVID-19 on MRS™

In Figure 1, violin plots illustrate the distribution and probability density of the MRS™ at baseline and after the virtual addition of the five repurposed drugs (one at a time, alone or in combination) into PACE participants’ drug regimens. Our simulation results show that the addition of repurposed drugs significantly enhanced the median MRS^TM^ (by 2 to 7 points, *p* < 0.0001). The combination therapy of HCQ + AZ or CQ + AZ was associated with further increases of the MRS^TM^ compared to HCQ and CQ alone (median, 20 and 19 vs. 17 and 16, respectively; *p* < 0.0001).

The impact of repurposed drugs on the frequency distribution of participants allocated to Low, Moderate, and High MRS^TM^ groups is shown in Figure 2. The virtual addition of repurposed drugs was associated with a lower number of patients in the Low-risk group and a significant increase in the number of participants with a High MRS^TM^. Compared to the baseline situation, the percentage of participants in the High-risk group was augmented by 50%, 103%, 43%, 88%, and 120% with HCQ, HCQ + AZ, CQ, CQ + AZ, and LPV/r, respectively (*p* < 0.0001). A larger fraction of participants were moved to the High-risk group when the CQ + AZ or HCQ + AZ combination was added to their drug regimen compared to CQ and HCQ alone (*p* < 0.0001).

Figure 3 shows the effects of the virtual addition of repurposed drugs on individual aggregated risk factors including FAERS, CYP450 drug interaction burden, ACB, and SB burden scores. As mentioned previously, the FAERS score uses the FDA Adverse Event Reporting System to compute risk of adverse drug events (ADE) which is based on computation of a drug regimen relative odds ratio for ADE using the FDA pharmacovigilance database. The median FAERS score was significantly increased with the virtual addition of repurposed drugs to participants’ drug regimens. The increase observed in FAERS values was of a similar magnitude for all repurposed drugs tested, with probability density estimates greater for HCQ + AZ, CQ + AZ, and LPV/r. The CYP450 drug interaction burden score was significantly higher following the virtual addition of LPV/r and HCQ (regardless of whether it was alone or combined with AZ) (Appendix A). The virtual addition of CQ into patients’ drug regimen also slightly affected the CYP450 drug interaction burden (Appendix A). Both ACB and SB risk scores were not affected following the virtual addition of repurposed drugs.

### 3.3. Simulated Effects of Repurposed Drugs for COVID-19 on the LQTS Risk Score

The effects of virtually adding repurposed drugs on drug-induced LQTS risk scores was examined while performing additional simulations with terfenadine (a drug known to prolong the QT and cause torsade de pointes) and fexofenadine (used as negative control as fexofenadine is not associated with QT prolongation) [37]. As illustrated in Figure 4, all repurposed drugs and terfenadine demonstrated a significant increase in the drug-induced LQTS score. No change in LQTS score was observed following the virtual addition of fexofenadine. Our simulations showed that the addition of HCQ and CQ alone produced changes in the distribution of the drug-induced LQTS score to an extent similar to the one observed with terfenadine. The magnitude of changes with LPV/r was also comparable to the one observed with terfenadine. The HCQ, CQ, and terfenadine drugs combined with AZ were associated with the highest increases in the drug-induced LQTS risk score compared to baseline (Appendix A and Figure 4).

Figure 5 illustrates the number of participants with Moderate and High risk of drug-induced LQTS after virtual addition of repurposed drugs to their drug regimens. Percentage of PACE participants in the High drug-induced LQTS risk group increased by 81%, 192%, 81%, 189%, 89%, 89%, and 192% with HCQ, HCQ + AZ, CQ, CQ + AZ, LPV/r, terfenadine, and terfenadine + AZ. No change in the frequency distribution was observed following the addition of fexofenadine.

The contribution of sex as a covariable was evaluated during the LQTS simulation process. Figure 6 shows violin plots for the drug-induced LQTS risk score distribution and probability density estimates stratified by sex. Our results indicate that those drugs affecting the drug-induced LQTS risk have greater effects on females than on males (*p* < 0.05). The effect of size was assessed, and the analysis indicates that while all female–male differences are statistically significant, the addition of repurposed drugs or terfenadine, except for fexofenadine, increased the difference between females and males compared to the baseline (*p* < 0.0001).

In our studied population, commonly used drugs known to prolong the QT interval with a clear association with the induction of torsade de pointes were donepezil, escitalopram, and citalopram: more than 20% of PACE participants were taking one of these medications. The most commonly prescribed drugs contributing to the drug-induced LQTS risk are listed in Table 2.

## 4. Discussion

The most vulnerable patients at risk for serious illness with COVID-19 are the older adults and individuals with hypertension, diabetes, cardiovascular diseases, chronic respiratory disease, and cancer [34,45,46]. Hence, PACE participants considered in our study are at increased risk since they are, on average, 76 years old and have common chronic conditions such as hypertension, type 2 diabetes, and COPD. Simultaneous use of multiple drugs, as observed in our PACE cohort (with a mean of 11.8 different drugs per day), is associated with an increased risk of multi-drug interactions and of ADE. The risk of using any of the proposed repurposed COVID-19 drugs in such an elderly population with polypharmacy has not been specifically evaluated in any prospective randomized clinical trials—this population is underrepresented in COVID-19 clinical trials. Although 80% of COVID-19-related deaths occur among people aged 65 and over [47], a recent review led by Harvard Medical School researchers showed that older adults are routinely excluded from clinical trials on COVID-19 treatment [48]. Similarly, in a recent randomized trial investigating whether HCQ could reduce COVID-19 in adult outpatients, the older population with comorbidities was underrepresented; the median age was 40 years, and 68% of participants reported no chronic medical conditions [49]. In addition, an exhaustive list of medications associated with potential QT prolongation was excluded, and such exclusion criteria would contribute to underestimating the risk of cardiac side effects in patients with polypharmacy. In search of benefit, one cannot overlook risk. Our approach represents a powerful strategy to estimate risk associated with drugs without exposing patients to potential ADE, including death.

We have demonstrated that the total MRS^TM^ of PACE participants’ drug regimens was hypothetically increased following the virtual addition of any of all tested COVID-19 repurposed drugs. In older adults with polypharmacy, this hypothetical increase in MRS^TM^ was primarily associated with an increase in CYP450 drug interaction burden and drug-induced LQTS scores. The MRS^TM^ has been previously evaluated and validated in a sub-cohort of PACE participants as a medication risk prediction tool for ADE and medical outcomes including hospitalization, emergency department visits, and medical expenditure [21]. Bankes et al. reported that each point increase in the MRS^TM^ was significantly associated with an increase of ADE (OR = 1.09), additional emergency visits (+3 visits/100 patients/year), and an extra yearly medical expenditure of $1037. A difference of 2 to 7 points in the median value of the MRS^TM^ was observed in our study when repurposed drugs were virtually added to drug regimens. Even though a short treatment duration with repurposed drugs is considered, the anticipated consequences remain substantial. Further pharmacoeconomic analysis with prospective data will provide more accurate estimates.

The COVID-Safer study exploited a similar strategy by using retrospective data from hospitalized older adults with polypharmacy included in the primary MedSafer study, and they theoretically exposed this patient population cohort to a treatment of HCQ (5 days at a minimum dose of 600 mg daily) [50]. They identified possible drug–drug interactions as well as potential harmful outcomes such as increased toxicity of HCQ, risk of QTc prolongation or malignant cardiac arrhythmia, or risk of other adverse drug events requiring closer monitoring during therapy. Their cohort of patients was smaller including only 1001 unique patients compared to this current study. They identified that 59% (590) were receiving one or more medications that could potentially interact with HCQ, and of those, 43% (255) were flagged as potentially inappropriate by the MedSafer tool. In agreement with COVID-Safer study, a simulation strategy can be used preemptively to identify older patients with polypharmacy; these patients are often ineligible for therapeutic trials or treatment with medication under investigation at high risk of drug–drug interactions. Consequently, the frequency of potential ADEs is underestimated.

While several aggregated factors are used to calculate the MRS^TM^, the significant increase in patients’ MRS™ after addition of repurposed drugs were mainly related to drug-induced LQTS and CYP450 drug interaction burden scores. No change in both the ACB and SB scores was observed: this finding was expected since these drugs are not known to show any pharmacologic properties on these risk factors.

The number of participants at high risk of drug-induced LQTS was significantly increased with the addition of any of the tested repurposed drugs. It is knowns that HCQ, CQ, and AZ can all prolong ventricular repolarization, as corroborated by studies raising concerns about the risk of arrhythmia and torsade de pointes with these drugs [11,12,13,14,15,16,51,52,53,54,55]. Based on the analysis of FAERS data, the relative odds ratio of QT prolongation/torsade de pointes for CQ, AZ, HCQ is 7.9, 2.6, 1.6, respectively. In April 2020, the FDA posted a drug safety communication against the use of HCQ and CQ for COVID-19 outside of hospital settings or clinical trials due to serious heart rhythms problems such as QT prolongation [9]. In the context of COVID-19, QT prolongation and risk of torsade de pointes have been identified and discussed for patients treated with HCQ or CQ, alone or with AZ: such observations support and validate results obtained through our simulation study [19,56,57,58,59]. Clinical trials in hospitalized COVID-19 infected patients have been terminated, or HCQ or CQ arms have been removed from their trials, for either safety or lack of efficacy issues [8,60].

Our results indicate that the virtual addition of repurposed drugs to a patient’s drug regimen significantly augmented the drug-induced LQTS risk score compared to the actual drug regimen. Notably, the drug-induced LQTS risk factor was of a similar magnitude to the one observed with terfenadine, a drug removed from the market because of drug-induced LQTS [61]. No difference was observed using fexofenadine as a negative control for drug-induced LQTS. This finding suggests that a simulation strategy based on a medication risk score represents a preemptive tool to predict patients at higher risk of drug-induced LQTS before exposing patients to a medication. Female sex is also a known risk factor for drug-induced LQTS as studies have shown that women have a greater susceptibility to torsade de pointes than men [62,63,64]. Our simulation approach confirmed that female sex was associated with higher drug-induced LQTS risk score than male.

The incidence of drug-induced LQTS/torsade de pointes is low, and it remains difficult to estimate an accurate incidence number since cases of torsade de pointes often are not recognized or are unreported [65,66]. In the MRS^TM^, the High-risk, drug-induced LQTS category corresponds to a score for drugs known to prolong the QT interval and associated with torsade de pointes. The combination of certain conditions such as drug interactions with QT-prolonging drugs, drugs causing electrolyte disturbances, hypokalemia, and older age can lead to a High drug-induced LQTS risk score. At baseline, the prevalence of patients in the High drug-induced LQTS score group was 0.29%; this number increased to 0.54% when HCQ, CQ, LPV/r, or terfenadine were added to participants’ drug regimens. When HCQ, CQ, or terfenadine were combined with AZ, the simulation showed a 3-fold increase in the prevalence (0.87%) of drug regimens with a High drug-induced LQTS score.

HCQ and CQ (alone or combined with AZ) are the oral repurposed COVID-19 therapeutic agents most studied in on-going clinical trials, followed by LPV/r. [67] As these medications are metabolized by various CYP450 isoforms, they carry the risk of causing or suffering from drug–drug interactions. The combination of LPV/r was associated with the highest increase in the CYP450 drug interaction burden score. It is well known that ritonavir has the potential to significantly interact with many medications: ritonavir can competitively inhibit the metabolism of CYP3A substrates and induce numerous CYP450 isoforms such as CYP1A2, CYP2B6, CYP2C9/19, and UGTs [41,43]. Consequently, ritonavir, as a perpetrator, has the potential to cause several drug interactions. Conversely, HCQ and CQ, which exhibit a low affinity for the isoforms involved in their metabolisms, will suffer from drug interactions (competitive inhibition) acting as a victim drug. Approximately 7% of our studied population received CYP2C8 medications that can affect HCQ and CQ exposure. Although HCQ is mainly metabolized by CYP2C8, together, CYP2D6 and CYP3A4 contribute to ~30% of its metabolic clearance. More than 50% of patients were simultaneously taking CYP3A4 and CYP2D6-interacting drugs which can impede one-third of the HCQ elimination pathway.

## 5. Conclusions

Our study used a simulation strategy based on a medication risk score assessment used clinically to improve drug safety and reduce risks of ADE in people with underlying medical comorbidities and polypharmacy. Besides the current repurposed drugs tested, our strategy can be applicable to any new drug being proposed to be used for COVID-19 or other clinical situations. As we do not know the benefit associated with some drugs, our approach allows for the estimation of risk in patients with polypharmacy.

## 6. Patents

The MedWise Risk Score^TM^ (MRS^TM^) used in the current study is subjected to two patent applications: WO2019089725 (Population-Based Medication Risk Stratification and Personalized Medication Risk Score) and WO2017213825 (Treatment Methods Having Reduced Drug-Related Toxicity and Methods of Identifying the Likelihood of Patient Harm from Prescribed Medications).

In patent WO2019089725, embodiments of the invention relate to a system and method for population-based medication risk stratification and are used for generating a personalized medication risk score. The system and method may pertain to a software that relates pharmacological characteristics of medications and patient’s drug regimen data into algorithms that (1) enable identification of high-risk patients for ADEs within a population distribution and (2) allow computation of a personalized medication risk score which provides personalized, evidence-based information for safer drug use to mitigate medication risks. In brief, embodiments of the invention include algorithms that look at multiple factors that influence a medication regimen’s likelihood of causing a negative health effect. The following factors are used to drive the software’s algorithms to determine risk in respect to patient’s medication regimen: a relative odd ratio for a drug regimen risk of ADE based on the FDA Adverse Event Reporting System, the indices of anticholinergic burden, the indices of sedation effects, the risk of QT-interval prolongation, and the competitive inhibition on specific CYP450 isoforms by the drug regimen. The combinatorial assessment of these individual risk factors provides a comprehensive approach to medication risk stratification at a population level, as well as the possibility of personalized medication risk mitigation at the individual level by interpreting a Personalized Medication Risk Score. Hence, the output of this assessment is a quantitative score that can be used to measure and stratify risk of the occurrence of ADE due to a particular medication regimen. This quantitative score also allows the identification of patients at higher risk for multi-drug interactions within a population and thus require medication risk mitigation more so than others. This identification ability is of high importance for care providers who seek to know which of their patients require immediate attention. Once these patients are recognized, the software tool provides a personalized snapshot of the risk factors described above, empowering the provider to mitigate their medication risk accurately and efficiently.

To ensure the accuracy of the invention, the scoring mechanisms have been validated against literature and clinical cases. The software has been applied to various healthcare population settings numbering more than 30 million patients. In these applications, various high-risk groupings were identified and criteria were generated for patients of the highest risk through statistical aggregation. The tool has been found to typically identify the top 15 to 20 percent of high-risk patients for each risk factor as well as identify the approximate top 5 to 15 percent of the population considered at highest risk. Not only were high-risk members of the population identified, but the embodiments of the invention generated personalized medication risk score snapshots, which empower health professionals to generate recommendations to mitigate the established risks. The description of this Medication Risk Score™ was first published by Cicali et al. and is currently used in the course of the Centers for Medicare and Medicaid Services (CMS) Enhanced Medication Therapy Management (EMTM) Pilot program with MedicareBlueRx [23,32]. Complete description of algorithms and methodology pertaining to this patent can be found at https://patentscope.wipo.int/search/en/detail.jsf?docId=WO2019089725&tab=PCTDESCRIPTION (N = 190 claims, 17 drawings).

Patent WO2017213825 describes methods of determining whether specific drugs or patients carry an increased risk of causing or developing, respectively, long QT syndrome, or torsade de pointes and methods of treating such patients. The inventions provide a comprehensive approach that includes a number of factors that may influence a medication’s likelihood of causing drug-induced LQTS and/or torsade de pointes. In some embodiments, the Long QT-JT Index considers the scenario where a medication has the greatest chance of causing torsade de pointes—when a medication is the most “risky.” The following factors may be considered specific to each medication: (1) IC50 for block of I_Kr_; (2) IC_50_ for block of I_Ks_; (3) IC_50_ for block of Navl 5 (sodium) current; (4) IC_50_ for block of Cavl.2 (calcium) current; (5) Inhibition of hERG trafficking; (6) Cmax of the drug at a test Dose; (7) Maximum daily dose of the drug according to labeling; (8) Protein binding of the drug; and/or (9) Drug–drug interaction coefficient (DDIC). In some embodiments, the DDIC considers the pharmacokinetics of the medication, i.e., whether it has a high extraction ratio (low bioavailability) or low extraction ratio (high bioavailability), relative CYP450 enzymatic pathways involved in the clearance of the drug. While the medication-specific risk index will be helpful when scrutinizing a single medication, the reality is that patients take many medications and have individual risk factors that may predispose or protect them from QT prolongation and torsade de pointes (TdP). In some embodiments, a patient-specific Long QT-JT Score is provided that is dynamic based on the patient’s current conditions and concomitant medications.

The scoring mechanisms described in this patent have been validated against literature cases of known TdP. More than 50 cases of documented TdP have been identified due to medication use and/or medication interactions, and their risk score has been calculated based on the methods described. In such cases, the risk scores are generally above 10. The patent description contains 236 claims and 22 drawings.

## Figures and Tables

**Figure 1 jcm-09-02591-f001:**
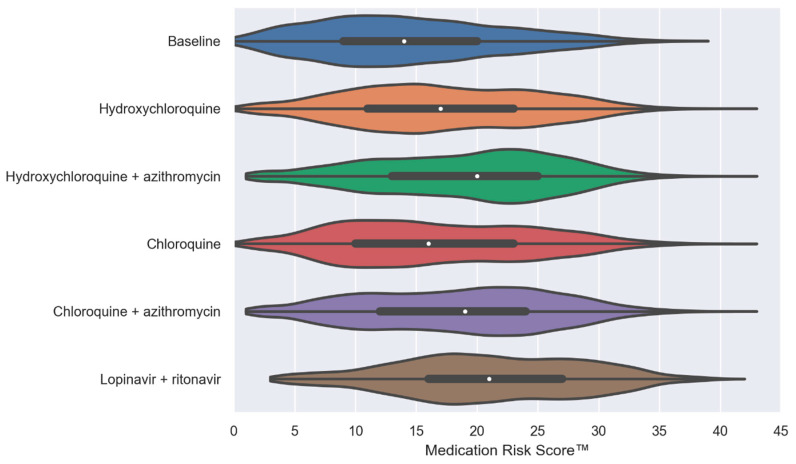
Violin plots of the MedWise Risk Score™ at baseline and after the virtual addition of repurposed drugs into Programs of All-inclusive Care for the Elderly (PACE) participants’ drug regimens. The white dots are the medians, and the colored areas are probability density estimates. *p* < 0.0001 compared to the baseline for all repurposed drugs. Abbreviations: HCQ, hydroxychloroquine; AZ, azithromycin; CQ, chloroquine; and LPV/r, lopinavir + ritonavir.

**Figure 2 jcm-09-02591-f002:**
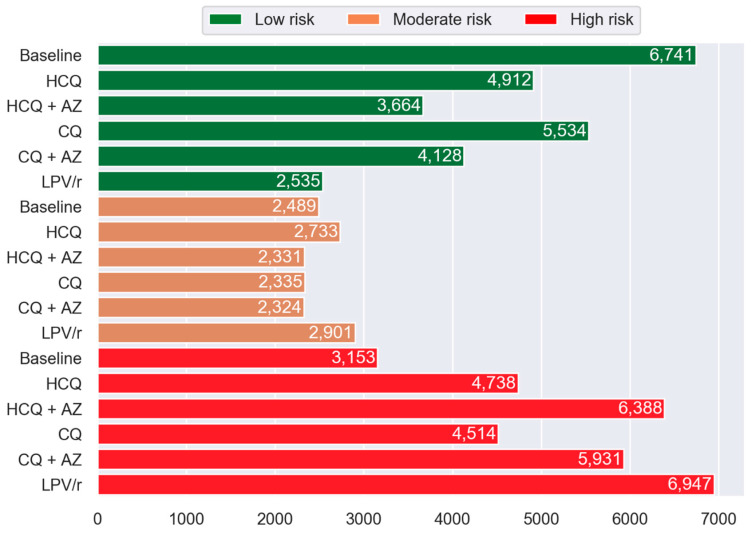
Histogram showing the frequency of PACE participants in Low, Moderate, and High MRS™ categories by drug compared to the baseline. The green, orange, and red represent the Low-risk (MRS^TM^ 0 to <15), Moderate-risk (MRS^TM^ ≥ 15 to <20), and High-risk (MRS^TM^ ≥ 20) group categories, respectively. Abbreviations: HCQ, hydroxychloroquine; AZ, azithromycin; CQ, chloroquine; and LPV/r, lopinavir + ritonavir.

**Figure 3 jcm-09-02591-f003:**
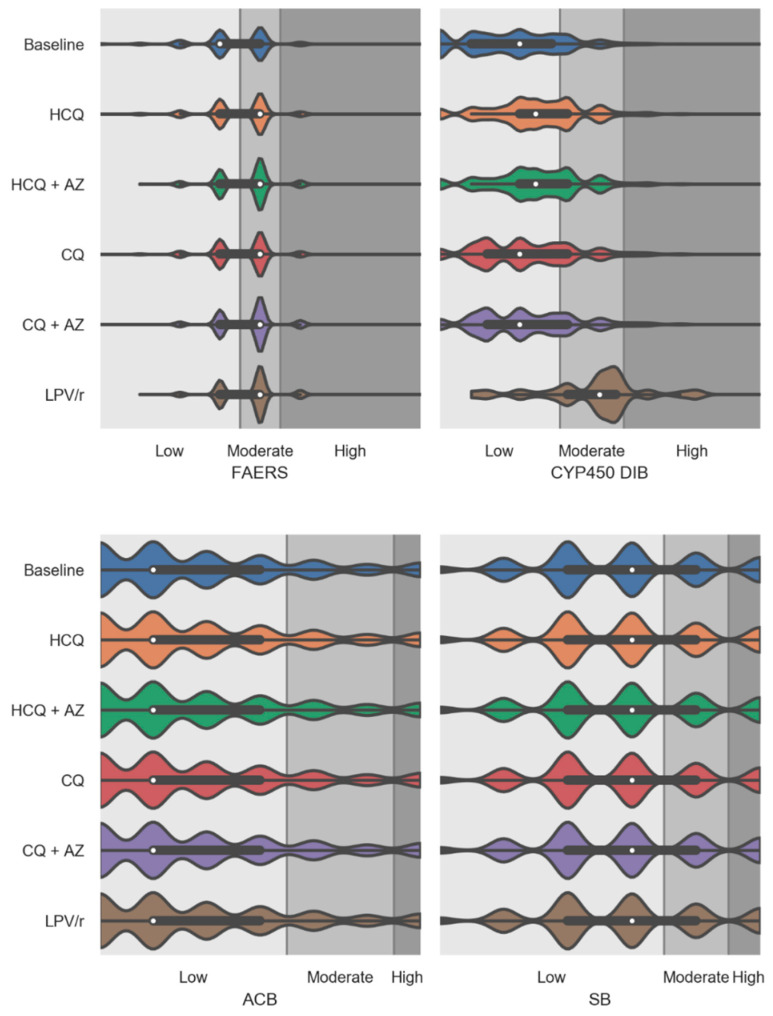
Violin plots showing the medication risk scores for individual factors including FDA Adverse Event Reporting System (FAERS), CYP450 drug interaction burden, anticholinergic burden, and sedative burden scores at baseline and after the virtual addition of repurposed drugs into PACE participants’ drug regimens. The white dots are the medians, and the colored areas are probability density estimates. Abbreviations: HCQ, hydroxychloroquine; AZ, azithromycin; CQ, chloroquine; and LPV/r, lopinavir + ritonavir.

**Figure 4 jcm-09-02591-f004:**
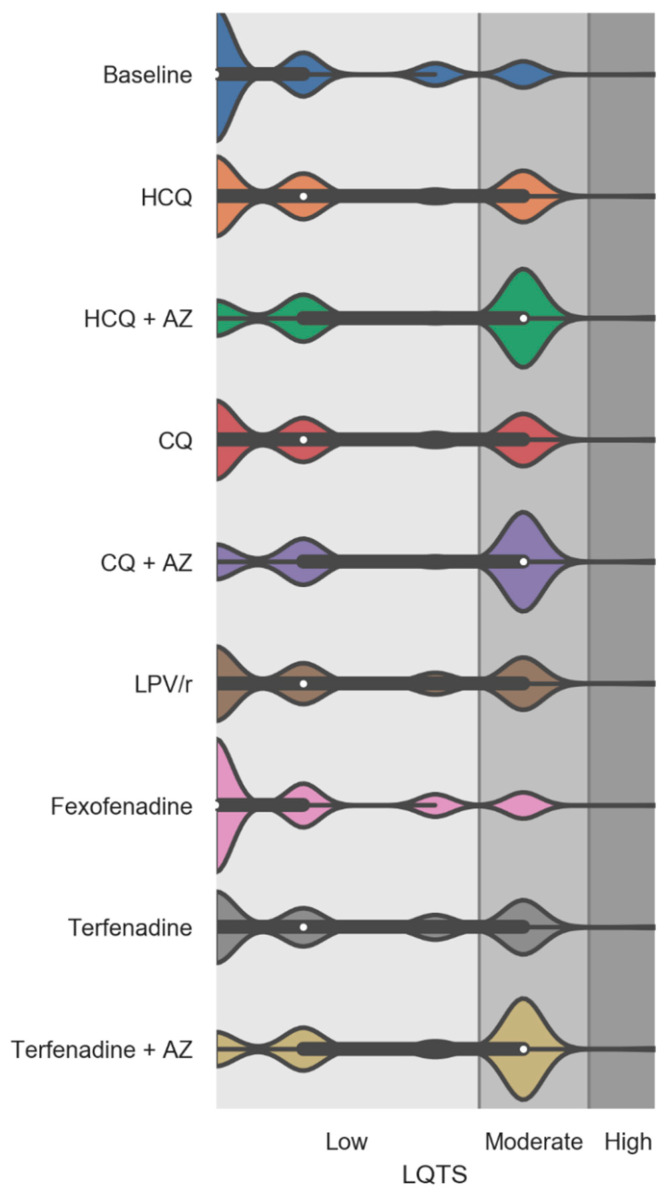
Violin plots showing the drug-induced Long QT Syndrome (LQTS) risk score factor at baseline and after the virtual addition of repurposed drugs into PACE participants’ drug regimens. Fexofenadine was used as a negative control and terfenadine alone and combined with AZ were used as positive controls on drug-induced LQTS score. The white dots are the medians, and the colored areas are probability density estimates. *p* < 0.0001 compared to the baseline for all repurposed drugs and terfenadine (alone and combined with AZ). Abbreviations: HCQ, hydroxychloroquine; AZ, azithromycin; CQ, chloroquine; and LPV/r, lopinavir + ritonavir.

**Figure 5 jcm-09-02591-f005:**
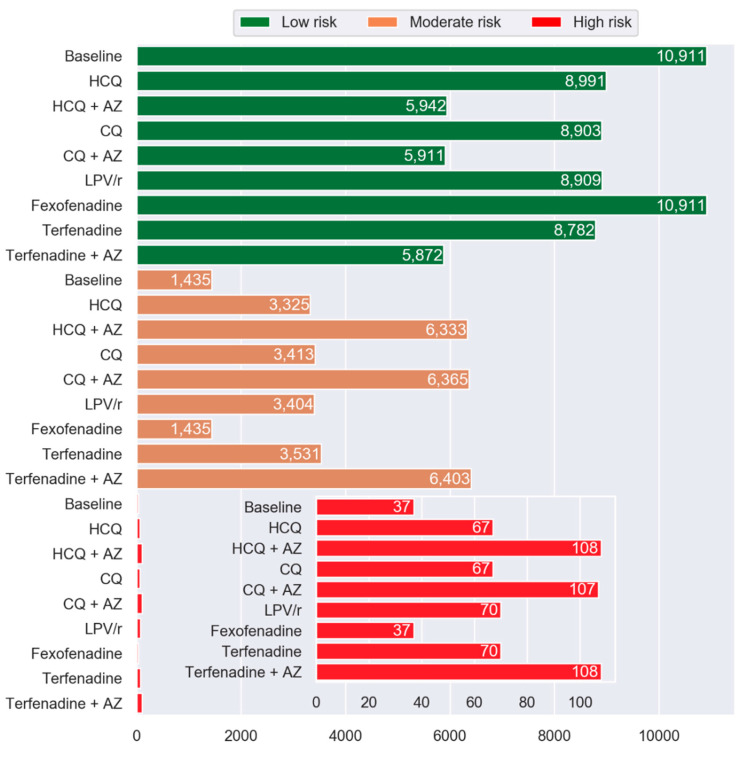
Histogram showing frequency of PACE patients in Low, Moderate, and High LQTS risk categories by drug compared to the baseline. The inset shows the High-risk categories. The green, orange, and red represent the Low-risk, Moderate-risk, and High-risk, respectively. Abbreviations: HCQ, hydroxychloroquine; AZ, azithromycin; CQ, chloroquine; and LPV/r, lopinavir + ritonavir.

**Figure 6 jcm-09-02591-f006:**
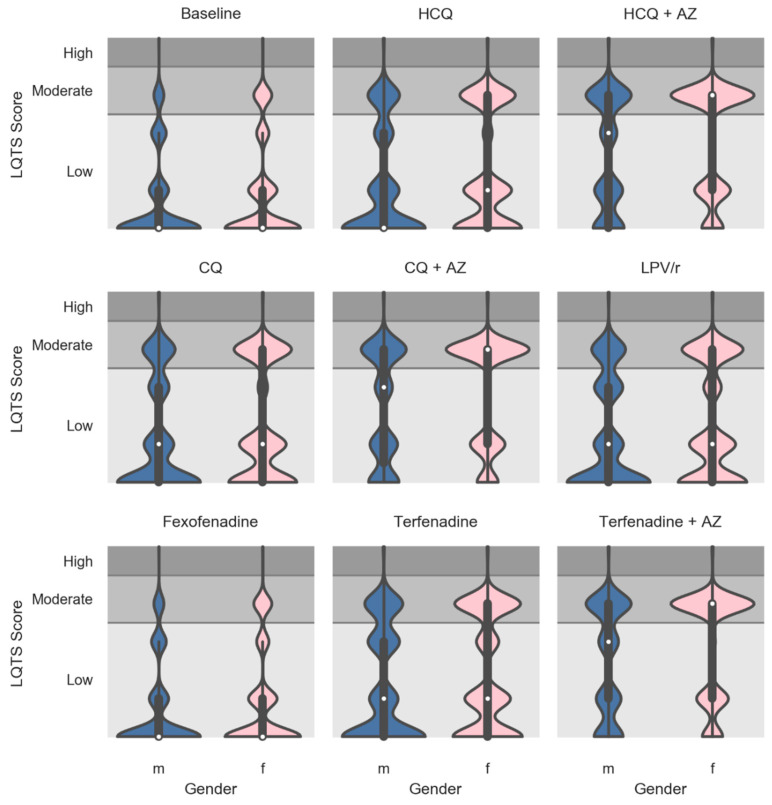
Violin plots showing the drug-induced LQTS risk score factor broken down by sex at baseline and after the virtual addition of repurposed drugs into PACE participants’ drug regimens. Fexofenadine and terfenadine (alone and combined with AZ) were used as negative and positive controls, respectively, on drug-induced LQTS scores. The white dots are the medians, and the colored areas are probability density estimates. Male and female are represented by blue and pink colors, respectively. *p* < 0.05 female vs. male for the baseline and for all repurposed drugs and controls. Size of effect, *p* < 0.0001 for all added drugs vs. the baseline, except for fexofenadine. Abbreviations: HCQ, hydroxychloroquine; AZ, azithromycin; CQ, chloroquine; LPV/r, lopinavir + ritonavir; m, male; and f, female.

**Table 1 jcm-09-02591-t001:** Demographic characteristics of the studied population.

Characteristics at Baseline	
Total number of patients, *n*	12,383
Age (years); mean ± SD (range)	76 ± 10 (55 to 89)
Sex; male, N (%) female, N (%)	4039 (32.6%) 8344 (67.4%)
Number of drugs/day per patient; N ± SD (range)	11.8 ± 5.7 (0–47)
Total MRS^TM^; mean ± SD (range)	14.4 ± 7.7 (0–39)
- Low-risk group *; N (%) mean ± SD	6741 (54.4%) 8.6 ± 3.7
- Moderate-risk group *; N (%) mean ± SD	2489 (20.1%) 16.9 ± 1.4
- High-risk group *; N (%) mean ± SD	3153 (25.5%) 24.8 ± 3.7
Patients currently receiving prescribed drugs proposed for repurposing: - HCQ, N (%) - CQ, N (%) - AZ, N (%) - LPV/r, N (%)	106 (0.9%) 0 125 (1.0%) 5 (0.04%)
Potential CYP450 drug-drug interactions, patients currently receiving: - CYP2C8 inhibitors or competitive substrates ^ŧ^, N (%) - CYP2C8 inducers ^ŧ^, N (%) - CYP3A4 inhibitors or competitive substrates ^ŧ^, N (%) - CYP2D6 inhibitors or competitive substrates ^ŧ^, N (%) - Combination of CYP3A4 + 2D6 inhibitors or competitive substrates^ŧ^, N (%) - Combination of CYP2C8 + 3A4 + 2D6 inhibitors or competitive substrates ^ŧ^,N (%)	693 (5.6%) 137 (1.1%) 8952 (72%) 7439 (60%) 5861 (47%) 415 (3.4%)

* MRS^TM^ risk groups are defined as follows: MRS^TM^ values < 15 are classified into the Low-risk group, MRS^TM^ values ≥ 15 to <20 into the Moderate-risk group, and MRS^TM^ values ≥ 20 into the High-risk group. ^ŧ^ Competitive substrates are drugs exhibiting a high affinity for a respective isoform. Concomitant administration of these drugs with the weak-affinity substrates HCQ and CQ is associated with potentially significant drug interactions. LPV is also a weak substrate of CYP3A4, but since its administration was always with ritonavir, a strong inhibitor of CYP3A4, no additional inhibition and interaction were considered for LPV. However, ritonavir, when virtually added to a patient regimen, was considered as a perpetrator drug. ASA, acetylsalicylic acid; AZ, azithromycin; COPD, chronic obstructive pulmonary disease; CQ, chloroquine; HCQ, hydroxychloroquine; GERD, gastro-esophageal reflux disorder; LPV/r, lopinavir boosted with ritonavir; MRS^TM^, medication risk score; SD, standard deviation.

**Table 2 jcm-09-02591-t002:** Common prescribed drugs in our cohort of patients that can contribute to drug-induced LQTS/torsade de pointes. Risk categories depend on whether drugs can cause QT prolongation or torsade de pointes: the Long QT-JT index used to derive the LQTS risk score and CredibleMeds are presented.

Drug Names	Number of Patients, N (%)	Long QT-JT Index Category *	CredibleMeds Category ^ŧ^
Furosemide	2629 (21%)	Moderate	Conditional
Pantoprazole	2402 (19%)	Moderate	Conditional
Sertraline	1583 (12%)	Low	Conditional
Trazodone	1479 (11%)	High	Conditional
Famotidine	1422 (11%)	Moderate	Conditional
Omeprazole	1395 (11%)	Moderate	Conditional
Hydrochlorothiazide	1359 (10%)	Moderate	Conditional
Donepezil	1073 (8%)	Moderate	Known
Mirtazapine	1004 (8%)	Moderate	Possible
Quetiapine	853 (6%)	High	Conditional
Escitalopram	752 (6%)	High	Known
Citalopram	681 (5%)	High	Known
Risperidone	500 (4%)	High	Conditional
Mirabegron	429 (3%)	Low	Possible
Aripiprazole	400 (3%)	Moderate	Possible
Venlafaxine	380 (3%)	High	Possible
Fluoxetine	357 (2%)	Moderate	Conditional
Olanzapine	270 (2%)	Low	Conditional
Esomeprazole	216 (1%)	Moderate	Conditional
Amiodarone	212 (1%)	High	Known
Ondansetron	177 (1%)	High	Known
Paroxetine	182 (1%)	Moderate	Conditional
Loperamide	154 (1%)	High	Conditional

* Long QT-JT Index values ≤ 15 are associated with High risk of drug-induced LQTS, values 16–100 are associated with a Moderate risk, and values 101-999 are associated with Low risk. Patient-specific risk (Long QT-JT Score) is determined by using the drug-specific Long QT-JT Index combined with other factors such as age, sinus rhythm, use of diuretics, use of Class 1A,1C or III antiarrhythmic drugs, magnesium and potassium levels, and QT interval value [30]. ^ŧ^ CredibleMeds categories are described as follows: Known risk of torsade de pointes is drugs that prolong the interval and are clearly associated with a known risk of torsade de pointes. Conditional risk is drugs associated with torsade de pointes but only under certain conditions of their use or creating conditions that facilitate or induce torsade de pointes. Possible risk is drugs that can cause QT prolongation but have no clear evidence for a risk of torsade de pointes [44].

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
