# Peer review of "Risk of Adverse Drug Events Following the Virtual Addition of COVID-19 Repurposed Drugs to Drug Regimens of Frail Older Adults with Polypharmacy"

_jcm, 2020, doi:10.3390/jcm9082591_

Round 1

Reviewer 1 Report

Abstract:

Missing a word before *strategy;

Would remove the term innovative as this concept is not new.

Introduction:

May mention that the EUA for HCQ has been revoked at this time;

Suggest not to use the word “cure” but rather may say only two drugs in the hospitalized setting have shown some promise (remdesivir and dexamethasone- REF NEJM recently published paper) but that in the community we have yet to find a treatment that decreases symptoms or the rate of severe complications

Re safety of hydroxychloroquine would add that there is a risk of QTc prolongation and that an FDA warning was released to this point. And that in hospitalized patients several trials have been stopped for futility or failure to show benefit (Solidarity and Recovery- both not yet published).

MRS is associated with X, Y and Z-> could this be rephrased to say a higher MRS is associated with (if I have interpreted correctly)

Again, would remove the term innovative

Methods:

I know these are defined in the intro: FAERS, ACB, SB,

But they are not common acronyms and I had to go back and look them up

Prefer these be spelled out for easier comprehension of the methods (in general the manuscript already has quite a few acronyms for the reader to work through).

How were sub-categories for low medium and high risk determined?

What are the upper and lower bounds of the MRS? How low can it be and how high can it go? And what are the cut-points to correspond to low, medium and high risk? And what does a low, medium and high risk correspond to in terms of risk of an ADE and over what timeframe?

E.g. a low risk score is less than 10 and corresponds to a <5% risk of ADE over the following six months (some description like this for each of the categories would be really helpful)

I also found the cut point data listed as a footnote in Table 1 but I think this needs to be moved to the main text.

Results+ Figures/Tables:

Do you capture atrial fibrillation? Surprised this was not more common

Could you specify that the drugs listed increase HCQ concentrations (rather than state that they affect the levels)

In table 1, normally we would list each disease and each drug in a separate row and include the N(%) for each rather than list common diseases and drugs as a conglomerate without proportions.

In the section where CYP inducers/inhibitors are listed the alignment of the table is off I think

This probably needs to be fixed : In Error! Reference source not found., (line 180 page 7)

In Table 2 would present the mean or the median but not both

Could probably take out the column for p-values and add a * where significant

Figures are very nice and interesting but similar results are presented in both figure form and table form. Probably would pick one way to present the data and then put the other format in the supplement.

I’m not very familiar with FAERS and I think others may not be as well; in the footnotes in the tables would remind the reader that this is a risk of ADE based on a calculated OR from pharmacovigilance database and would spell out the acronym as you have for sedative score and anticholinergic score.

I think the terfenadine and fexofenadine positive and negative control data could be in a supplement. It’s also not totally clear to me why these were required for the QT prolongation simulations as opposed to simulations for, say, anticholinergic burden. Why does only this one subcategory of ADE require positive and negative controls?

Would pick either histogram or violin plot; no need to include both in the main paper.

M and F should be spelled out in Figure 6 as male and female

Would move some of the figures or histograms to the supplement and instead bring this info into the main paper: prescribed drugs contributing to the drug-induced LQTS risk are listed in the Supplementary Table 277 S4.

For clinicians, it would be important to know the most common drugs that interact, not merely that drugs that interact exist. A lot of the primary data presented shows that the addition of the repurposed drugs worsens the risk for ADE in combination with baseline drugs, but it would be helpful to know readily (ie in the body of the main paper) what drugs are most commonly interacting that patients are taking at baseline, but this appears a mere mention at the end of the results section.

Discussion:

Would change elderly to older adults throughout

Re this statement: The risk of using any of the proposed repurposed COVID-19 drugs in such an elderly 286 population with polypharmacy has not been evaluated in any prospective randomized clinical trials

I think it would fairer to say that this population is underrepresented in clinical trials

That said, they will make up a population of patients in the hospitalized trials (recovery and solidarity); it may be for this very reason that HCQ for example is not beneficial in hospitalized patients as many of the participants in the hospital will be older, sicker, frailer, with polypharmacy. This may make the balance of harm: benefit favour harm for hospitalized patients.

Certainly, in the community trials that have been published in the last week (Boulware trial Annals of internal medicine for example) this population is not at all represented. Would update paper with this reference and nuance the above discussion with in-hospital data vs community data.

Remove innovative in discussion. It’s not needed.

Suggest that you contextualize this study with the very similar study that already exists  (Ross et al JAGS, COVID-SAFER) which did something very similar to this study and compare contrast the approaches. Could decrease the discussion of HCQ/CQ and focus a bit more on LPA/R given COVID-SAFER focused on HCQ. A lot of the concepts you discuss are in that paper so I think it would be worthwhile to read it and then advance the discussion a bit further in your manuscript.

Table S3- would add which of the repurposed drugs this represents an interaction with

Reviewer 2 Report

This manuscript details the potential risk of proposed COVID treatments to older adults with polypharmacy and many comorbidities.  The authors should be commended on their work. 

Abstract:

Line 15: is mandatory for what?  Mandatory for the new indication and population?  If a drug is on the market, there is an assumption that a risk benefit assessment has been done, so please add more to your sentence to give better context.

Introduction:

Line 35: Please update the COVID data as you can to ensure the most recent data is available.

Line 37: HCQ and CQ had received (it is no longer present tense).  As of 6/15/2020 the FDA revoked the EUA. 

Line 41: ... safety of administering... 

Line 52: What is the association (positive or negative) with the outcomes?

Line 64: LPV/r was originally thrown out but has not provided any efficacy for COVID.  May not be worth reporting in the manuscript as the clinical use of LPV for this condition never materialized. 

Line 97: HCQ was often given in a number of different dosing regimens for COVID.  How did you select 400mg PO BID?  Dosing has ranged from 400mg/day to 1200mg/day so please justify why only one dose was chosen to be studied?  

Line 111: What point in time was used to capture the patient's current medication list? What is meant were filtered by dates?  Was that to ensure only current meds were include? Are only Rx included on this list?  Is there any OTC data available?

Table 1: thyroxine or levothyroxine?

Line 180: Please address "Error! Reference source not found"

Figure 2: This seems very confusing.  Is there a way to organize by baseline, HCQ, HCQ + AZ, etc so the read can see the shift for each drug as opposed to scanning line by line as it is presented now?

Table 2: Is the p-value comparing mean or median vs baseline?  Please clarify. 

Figure 5: Again, can this be reconfigured as it is hard to understand unless each drug is grouped together. 

Line 290: This is a potential increase in the MRS based on a hypothetical situation.  There is mention of virtual, but please be more explicit with the term hypothetical.  Also may be worth stating that the increase in MRS is due to the DDI and LQTS portions (not the ABS or sedation piece) to allow the reader to hone in on the direct risks to the older adult. (It is mentioned later in the paragraph, but earlier is a more direct way to convey that info). 

Line 306: high-risk
